# XR (Extended Reality: Virtual Reality, Augmented Reality, Mixed Reality) Technology in Spine Medicine: Status Quo and Quo Vadis

**DOI:** 10.3390/jcm11020470

**Published:** 2022-01-17

**Authors:** Tadatsugu Morimoto, Takaomi Kobayashi, Hirohito Hirata, Koji Otani, Maki Sugimoto, Masatsugu Tsukamoto, Tomohito Yoshihara, Masaya Ueno, Masaaki Mawatari

**Affiliations:** 1Department of Orthopaedic Surgery, Faculty of Medicine, Saga University, 5-1-1 Nabeshima, Saga 849-8501, Japan; takaomi_920@yahoo.co.jp (T.K.); h.hirata.saga@gmail.com (H.H.); masa2goo99@yahoo.co.jp (M.T.); tomohito4113@yahoo.co.jp (T.Y.); f8286@cc.saga-u.ac.jp (M.U.); mawatam@cc.saga-u.ac.jp (M.M.); 2Department of Orthopaedic Surgery, School of Medicine, Fukushima Medical University, Fukushima 960-1295, Japan; kojiotani1964@gmail.com; 3Innovation Lab, Teikyo University Okinaga Research Institute, Tokyo 173-8605, Japan; sgmt@med.teikyo-u.ac.jp

**Keywords:** augmented reality, extended reality, mixed reality, navigation, spine surgery, virtual reality, telemedicine

## Abstract

In recent years, with the rapid advancement and consumerization of virtual reality, augmented reality, mixed reality, and extended reality (XR) technology, the use of XR technology in spine medicine has also become increasingly popular. The rising use of XR technology in spine medicine has also been accelerated by the recent wave of digital transformation (i.e., case-specific three-dimensional medical images and holograms, wearable sensors, video cameras, fifth generation, artificial intelligence, and head-mounted displays), and further accelerated by the COVID-19 pandemic and the increase in minimally invasive spine surgery. The COVID-19 pandemic has a negative impact on society, but positive impacts can also be expected, including the continued spread and adoption of telemedicine services (i.e., tele-education, tele-surgery, tele-rehabilitation) that promote digital transformation. The purpose of this narrative review is to describe the accelerators of XR (VR, AR, MR) technology in spine medicine and then to provide a comprehensive review of the use of XR technology in spine medicine, including surgery, consultation, education, and rehabilitation, as well as to identify its limitations and future perspectives (status quo and quo vadis).

## 1. Introduction

Advances in computing power have led to a significant advancement of virtual reality (VR), augmented reality (AR), and mixed reality (MR) technologies. Recently, the term XR (extended reality) technology, which more broadly integrates VR/AR/MR technologies, has been the focus of much attention. VR is “an immersive, completely artificial computer-simulated image and environment with real-time interaction” [1]. It is widely used to reproduce the human body structure, pathophysiology, and clinical scenes with a sense of realism. VR in spine medicine has been used most often for medical education, surgical simulation and planning, and intraoperative guidance [2,3]. AR has been described as “the concept of digitally superimposing virtual objects onto physical objects in real space so that individuals can interact with both at the same time” [4]. In practice, computer-generated images are overlaid on real-world images and displayed on video projectors, computers, or tablets. MR, a hybrid of AR and VR, is the result of blending the physical world with the digital world [5,6] and has recently garnered attention, mitigating the limitations of VR’s exclusion of the real-world environment and AR’s inability to interact with three-dimensional (3D) data packets [7]. VR and MR have primarily been applied in teaching and preparatory roles, while AR is mainly applied in hands-on surgical settings. The use of MR in spine medicine allows the surgeon to access intraoperative information about the patient’s anatomy and superimpose virtual holographic elements on the superficial anatomy of the actual patient in real-time on the operating table, enabling holographic-based navigation [8,9]. XR is a general term encompassing VR, AR, and MR. XR refers to all real-and-virtual combined environments between human and computer-generated input processed to create an interactive environment. This review covers VR, AR and MR, but for the sake of clarity, they are all referred to as XR.

In recent years, with rapid advancement and consumerization of XR technology, the use of XR technology in spine surgery has also become increasingly popular. The rising use of XR technology in spine medicine can also be attributed to the recent wave of digital transformation, which was further accelerated by the COVID-19 pandemic and the increase in minimally invasive spine surgery (MISS), which we will discuss in detail in Section 2. Digital transformation involves several interesting technologies or devices, which are described in Section 3. According to a review of 8399 articles on VR and AR research published in the medical field from 1992 to 2020, which were analyzed using bibliometric methods, the most popular research topics were diagnostic procedures, surgical procedures, and rehabilitation [10]. In addition, numerous reports have shown the usefulness of XR technology in medical education [10,11,12,13]. Therefore, in Section 4, we will discuss the following four major areas of spine medicine in which the application of XR technology has been accelerating: education, medical examination, surgery, and rehabilitation.

Figure 1 illustrates the structure of the above relationship.

Lastly, in Section 5, limitations, future directions, and possibilities will be discussed.

Given these facts and trends, it is time to discuss how such recent advances will coverage technological and medical insights. In this study, we adopted the narrative review method, which allows us to organize and analyze the existing literature in the field of XR technology in spine medicine more extensively, flexibly, and comprehensively in comparison to a systematic review. To that end, many pivotal articles in peer-reviewed scientific journals were selected, which helped to identify key XR technologies in spine medicine. The purpose of this narrative review is to identify the accelerators of XR (VR, AR, MR) technology in spine medicine and then to perform a comprehensive review of the use of XR technology in spine medicine, including surgery, consultation, education, and rehabilitation, as well as to identify its limitations and future perspectives (status quo and quo vadis).

## 2. What Has Accelerated the Introduction of XR Technology in Spine Medicine?

The introduction of XR technology in spine medicine has been accelerated by the recent wave of digital transformation and further accelerated by the COVID-19 pandemic, and the increase in minimally invasive spine surgery.

### 2.1. Digital Transformation in Spine Medicine

Technological advancement in recent years has brought intelligent computing to nearly every industry. The digital transformation wave has also arrived in spine medicine [14,15], largely due to the development of the four digital platforms for handling high-performance big data: collection, communication, editing, and viewing. For example, especially in telemedicine, the combination of high-performance information (i.e., high-resolution CT, wearable sensors, 360° operative camera), high-performance information communication technologies, high-performance edited images (i.e., 3D medical images or holograms) created using artificial intelligence (AI), and high-performance viewing using a head-mounted display (HMD) has been useful for diagnostics and treatment decisions [16,17]. Thus, advancement in digital transformation combined with the development of intelligence technology for handling high-performance information collection, communication, editing, and viewing has further expanded the application of XR technology in spine medicine.

#### 2.1.1. Collection

The collection of high-performance information in medicine can be attributed to the following: (1) advances in image quality and digitization of CT, MRI, and ultrasound scans; (2) development of wearable sensors; (3) development of inexpensive, compact, portable, and high-performance video cameras; and (4) development of surface topography methods in pediatric scoliosis. Wearable sensors (i.e., smartphones) can be used to collect many kinds of data remotely [18,19]. Notably, action cameras or 360° video cameras, which providing an immersive and realistic experience, have been widely used in most areas of surgery and in various clinical education settings [20,21,22]. The use of surface topography in pediatric scoliosis has been welcomed for its accuracy and reduced radiation exposure [23].

#### 2.1.2. Communication

Communication technologies with high performance information, including fifth generation wireless system (5G) and wireless fidelity (Wi-Fi), have the advantages of high data rates and low latency. These are the basis for XR technologies, including other emerging technologies, such as the Internet of wearable sensors, big data, 3D medical images and holograms, cloud computing, and AI, which can combine organically with 5G [16,24]. Furthermore, 5G network-based telerobotic spine surgery has been performed due to the aforementioned digital transformation and progress in robot-assisted spine surgery [25].

#### 2.1.3. Editing

The use of AI in the editing of high-performance information is already essential for image reconstruction. Clinicians have widely used 3D medical images edited by AI and high-performance communication technology. These images vividly reproduce the human body structure, pathophysiology, and clinical scenes with a sense of realism. This information could support medical treatment, including navigation surgery, education of medical students and residents, patient explanation, rehabilitation, and telemedicine [26,27]. Three-dimensional medical images and holograms are more practical than conventional models such as a 3D-printed models, as they allow clinicians to move objects around or remove certain areas. They can also be easily obtained by inputting high-performance information.

#### 2.1.4. Viewing

The most common way to view 3D medical images edited with the above-mentioned high-performance information is with a high-resolution display, which can take the form of either a traditional monitor or a head-mounted display (HMD) [28,29,30]. When combined with a 4 K or 8 K ultra-high-definition monitor system, it can provide sharper and clearer streaming video, providing detailed content that resolves information beyond the retina and helps to make a visual diagnosis [16]. Recent improvements in the medical image analysis and visualization equipment have led to the use of 3D medical images and holograms in clinical practice [5]. With the evolution of HMD, XR technology has been integrated into HMD systems. Three-dimensional holograms can be displayed on HMD; VR with HMD has been used to educate and guide trainees in pedicle screw fixation and has shown greater accuracy in comparison to traditional teaching methods [31,32]. AR with HMD has been primarily used to facilitate intraoperative navigation/guidance in MISS [33,34]. With the introduction of AR/MR with HMD, an omnidirectional hologram is projected onto the surgeon’s field of view, allowing the surgeon to concentrate on the surgical field without returning to the monitor [7].

### 2.2. The COVID-19 Pandemic

The social-distancing guidelines in response to the COVID-19 pandemic have accelerated digital transformation, creating a context that will continue to drive innovation and technological adoption. Similarly, the COVID-19 pandemic has brought about significant changes in medical education for residents and students around the world, disruptions to medical education, a reduction in elective operations, and restrictions on physical participation in workshops or conferences [35]. Thus, the use of technology to maintain medical treatment and education has become more rapid and innovative than ever before. As a result, many healthcare organizations have been increasingly interested in XR technology: patient care and management, the education of residents and medical students (i.e., online lectures, remote access to teaching ward rounds by XR technology [36], preoperative planning and simulation, and remote rehabilitation (telerehabilitation) [17,36,37].

With the continuous development and advancement of the abovementioned digital transformation, tele-medicine has become an essential part of medical information technology construction. In addition to isolation from social contact and interaction by stay-at-home policies, medical students have suffered from depression [38]. Therefore, telemedicine may be important not only for patients but also for medical students. Until vaccines are available or herd immunity is achieved, a repeat of the COVID-19 pandemic is expected, and no significant reversal of the digital trends is expected in the post-pandemic environment. Thus, we need to continue to pay attention to XR technology.

### 2.3. Minimally Invasive Spine Surgery (MISS)

Over the past 20 years, there has been an explosion of new MISS methods, which often require new skills and tools [7,39]. Advances in XR technology have facilitated the development of new skills and tools in spine surgery. MISS has become a common technique, bringing many benefits to both surgeons and patients; however, these techniques rely heavily on indirect visualization and/or navigation guidance [40,41,42]. XR technology, which can visualize the anatomy and guide the surgeon as precisely as intraoperative navigation, has been implemented in the MISS field. On the other hand, the narrow surgical field in MISS and the difficulty in obtaining accurate spatial awareness during surgery are factors that hinder the education and acquisition of surgical skills. In addition, MISS surgery is associated with a high rate of radiation exposure because surgeons must rely on X-ray images to confirm the instrumentation accuracy [43,44].

It has been reported that the usefulness of XR technology in spine surgery includes high-precision surgery, reduced radiation, and a shortened surgical time.

Therefore, in MISS, XR technology has played an increasingly important role in education and treatment [39].

## 3. Technologies Supporting the Digital Transformation of Spine Medicine

As discussed in Section 2, many XR technologies and devices have supported digital transformation in spine medicine. The following is a supplementary description of some particularly interesting technologies and devices.

### 3.1. 3D Medical Images and Holograms

The areas of application of 3D medical images and holograms are diverse and promising. They include education, patient follow-up and informed consent, and surgical simulation/navigation.

#### 3.1.1. Image Capture in Medical Application Systems Using XR Technology

Conventional two-dimensional imaging modalities mainly include X-rays, CT, and MRI, which often require years of clinical experience and a high degree of spatial imagination for an accurate diagnosis. For medical students and residents, the lack of an accurate understanding of the three-dimensional positioning of the organs makes it difficult to assess preoperative images and understand surgical techniques. Case-specific 3D holograms can be used as a new educational tool to improve the competency of medical students and residents [44], and as a tool for patient education [45]. Since XR technology is computer-based, it can perform learning activities that would be impossible in the real world. For example, students can already use Anatomage TABLE™ to observe 3D dissection to the center of a cell in VR. In this way, software development can provide an engaging learning experience that allows for a deeper understanding of complex concepts [46].

Surgeons frequently require the real-time estimation of 3D data from 2D images, and the application of XR technology in spine surgery could facilitate this task, improving patient safety and surgical efficiency. Medical images in spine surgery—even 3D-reconstructed images—can only be viewed on a flat monitor, which can lead to inaccurate spatial perception. Therefore, the intraoperative use of 3D holograms with high spatial awareness is desirable, which can enhance the safety of the operation (Figure 2A). Furthermore, 3D holograms can be viewed and moved through easy gesture-handling, without monitors in any place, including the operative room. Elements such as the integration of touchless features provide the setting of a sterile environment (Figure 2B).

#### 3.1.2. 3D-Based Classification Analysis

For some diseases where an understanding of the 3D anatomy is important (i.e., scoliosis), 3D-based classification using AI has been reported [47,48]. The use of such 3D-based classification systems for scoliosis may provide more insight into pathology than conventional two-dimensional (2D)-based classification systems. While the clinical relevance and practical usefulness of the 2D perspective has been considered, the 3D perspective will give us a more solid understanding of the relevance, prognosis, or targeted diagnostic evaluation of the spinal structure phenotypes in relation to pain, function, and disability profiles [49]. It may be time to reconsider the 2D evaluation of what is essentially a 3D structure. Furthermore, with the advancement of XR technology, 3D reconstruction will become easier and quicker. Therefore, 3D-based classification of spinal diseases will become more common in the coming years.

#### 3.1.3. Surface Topography in Spinal Posture

Surface topography (e.g., formetric 4D from DIERS Medical Systems (Chicago, IL, USA)), which has a diagnostic accuracy comparable to that of X-ray, is being used increasingly frequently for postural analyses [50].

Particularly for scoliosis screening in children, surface topography is attracting attention because of its low radiation dose.

Screening is accompanied by problems such as the burden on doctors to read the images; however, with the ongoing advancement of AI, these problems will be solved [51].

### 3.2. Wearable Sensors

Smartphones can collect parameters that need to be measured in telemedicine, such as ECG, arrhythmia, falls, step accelerometer data, gyroscope data, temperature, blood saturation, pulse, respiration, sleep quality, falls, and walk assessments [50,51,52,53,54]. Wearable sensors, also known as the Internet of medical things (IoMT), can also be used to collect data in real time to help monitor and manage patients remotely [18,19]. Although telemedicine in orthopedics has lagged behind other specialties, such as radiology, cardiology, and psychiatry [55], there have been reports on the use of a combination of wearable sensors and VR for shoulder rehabilitation [56], the postoperative evaluation of lumbar disc herniation, smart shoes for lumbar spinal stenosis [52], and the monitoring of brace usage for scoliosis [57,58,59,60]. In terms of spine medicine, information on gait assessment by wearable sensors such as smartphones or smart shoes may be useful for the preoperative assessment of neurological gait disorders and postoperative monitoring of spine surgery [60].

With the COVID-19 pandemic, telemedicine using a combination of VR and wearable sensors has become popular [59,60]. Wearable sensors have the advantage of being able to more accurately monitor a patient’s gait and posture in “everyday” life, which would not be reflected in controlled tests performed in front of a doctor. Thus, wearable sensors have been widely used in various telemedicine fields, including patient education, medical examination, and rehabilitation. The combination of wearable sensors and telemedicine as an audiovisual communication platform using XR technology will bring many benefits to both healthcare providers and patients, including benefits in relation to cost, time, and healthcare accessibility for patients in remote areas.

### 3.3. Action Cameras and 360° Operative Cameras (Video)

The ideal camera for recording surgery should be small, lightweight, comfortable, easy to use, able to depict the surgeon’s view, provide high-definition images and video, have long battery life, be inexpensive, and allow for easy management of images and video [22]. Since such an ideal camera does not currently exist, in many cases, people cope by using different cameras for different purposes, or by using combinations of multiple cameras.

Action cameras that have been applied for surgical recording include the GoPro Hero Series (GoPro, Inc., San Mateo, CA, USA) [61,62] and Google Glass (Google, Inc., Mountain view, CA, USA) [22], which can collect blur-free video images from the surgeon’s perspective. The GoPro is a small, lightweight, high-resolution action camera that has mainly been used to record action sports, but which has recently been used in clinical education. When worn on a head-mount, it has a 149-degree wide-angle field of view and can capture most of the visual information available to the wearer [60].

Moreover, 360° operative video has been used not only to analyze surgical performance for medical education but also for orientation in new environments, team training, and formal multi-disciplinary examinations, because it allows users to view images from all angles at the same time, providing an immersive and realistic experience [20].

The high level of sterility required in spinal instrumentation surgery has restricted medical students’ access to surgery, and the COVID-19 pandemic has accelerated this trend. Moreover, the increase in the number of MISS procedures makes it more difficult for students to learn surgical techniques. In addition, conventional ceiling-mounted video camera images show medical students the anatomy in the image, but do not allow them to observe how the surgeon manipulates the surgical instruments. With the use of XR technology, high-resolution videos captured with an action camera from the surgeon’s point of view and 360° operative video have become more realistic and immersive through online live broadcasts of surgery, which may solve the abovementioned problems for medical students.

### 3.4. XR (VR/MR)-HMD

A series of recent studies have indicated that XR technology provides the opportunity to co-create experiences such as HMDs and goggle types of devices that can already produce high-quality graphics and experiences.

#### 3.4.1. Viewing via HMD

Three-dimensional holograms created before surgery are projected onto a clean surgical field during surgery and can be viewed in three dimensions, which improves spatial awareness and allows the images to be shared among the doctors involved in the operation [8,9]. There are two types of MR glasses on the market: HoloLens (Microsoft, Inc., Redmond, WA, USA) and Magic Leap1 (Magic Leap, Inc., Plantation, FL, USA) [7,63,64,65,66].

In recent years, compact, lightweight, and comfortable HMDs have become commercially available. These smart glasses can superimpose computer images translucently on images through the lens, allowing the surgeon to obtain anatomical and navigation information without narrowing their field of view. Furthermore, it will enable surgeons to proceed with surgery without taking their attention away from the procedure, resulting in shorter operative times and reduced radiation exposure [67,68,69]. In the field of neurospine surgery, several attempts to use smart-glass displays as intraoperative neuromonitors [70], endoscopic monitors [71], display devices for fluoroscopy [68], and 3D navigation screens [67,69] have been reported. Moreover, with VR-HMD as a video viewing device, the 360° operative images and surgeon view with action camera can provide residents and medical students with a sense of immersion and realism, allowing them to experience surgery in an educational way, rather than by just observing it [20,22].

#### 3.4.2. Recording via HMD

Some HMDs have camera and video recording capabilities, which may be a more useful adjunct to surgical education, information sharing, real-time consultation, and remote teaching and monitoring [28,29,30,72,73]. Surgical techniques have been passed down from generation to generation as tacit knowledge. Since it is difficult to quantify them, it is impossible to evaluate their quality, and it takes time to learn them. Operative videos from the surgeon’s perspective using HMDs, especially from experienced doctors, signify the digitalization of surgical techniques into digital content. Digitized surgical videos can be viewed anytime and anywhere with highly immersive and realistic VR-HMDs, enabling surgeons to provide highly accurate feedback, and medical students and residents to pass on tacit knowledge specific to surgery by reliving the surgery [39].

#### 3.4.3. VR-HMD as a Sensor

A cervical range of motion assessment is useful for preoperative and postoperative evaluation of the cervical spine, and for checking the effect of rehabilitation. VR systems have been shown to be convenient, non-invasive, harmless to the human body, and capable of assessing the cervical range of motion with a high degree of accuracy [72,73]. It could also be used to examine the relationship between data measured by VR equipment and clinical imaging data, and for follow-up applications in remote areas [72].

## 4. XR Technology in Spine Medicine: Education, Medical Examination, Surgery, and Rehabilitation

The four major fields of spine medicine where the application of XR technology has been accelerated are education, consultation, surgery, and rehabilitation.

### 4.1. Education

#### 4.1.1. The Effectiveness of Using 3D Holograms to Teach Anatomy to Medical Students and Residents

The dissection of cadavers is important in medical education, but there are ethical issues involved in using cadavers, as well as issues of time, cost, and availability of donated bodies for training. Thus, dissection using XR technology has been attracting attention.

Chytas et al. [74] provide an overview of the implementation of XR technology in anatomy education and recommend educators to incorporate XR technology, such as exposing details of human anatomy and surgical procedures in a VR environment and using AR. Naturally, 3D holograms help to improve the intuitive understanding and spatial awareness through stereoscopic vision [75]. Thus, 3D holograms edited with XR technology can effectively improve the quality of teaching, stimulate students’ interest in learning, and improve students’ understanding of anatomical knowledge. Beyond the improvement of anatomical knowledge, other advantages of XR technology that use 3D holograms for medical student anatomy education include 24 h free access and gamified elements, which contribute to increased satisfaction. The use of XR technology in anatomy education has been used in relation to internal human anatomy, the anatomy of the ear, nose, and temporal bone, surgery, neuroanatomy, and cardiac anatomy [10]. Thus far, there are no reports in the area of spine medicine.

In the context of spine surgery education and training, the use of XR technology, including 3D holograms for preoperative planning, and intraoperative visualization may promote a faster understanding of surgical anatomy and skills, accelerate the learning curve, contribute to accurate and safe surgery, and increase patients’ satisfaction [2,7,75,76,77,78].

#### 4.1.2. Tele-Education with XR HMD

Tele-education may also be beginning to gain prominence due to the COVID-19 pandemic. In medical education, HMD can provide distance-learning students with an immersive and involved experience that was not previously possible. Conversely, when teachers wore HMD (HoloLens 2) to educate medical students at a London teaching hospital and conducted educational ward rounds via remote access, the results were effective and highly satisfactory [36].

Mcknight et al. [39] reported that remote surgical guidance from an experienced surgeon using an iPad or Google Glass HMD as a display benefits trainees as well as experienced surgeons learning a new skill, such as shoulder arthroscopy or shoulder arthroplasty [79,80,81]. Thus far, there are no reports of tele-education in spine medicine, so this is a future issue.

### 4.2. Medical Examination

Telemedicine involves XR technology, including wearable sensors and high-performance video cameras, to allow healthcare providers to evaluate, diagnose, monitor, treat, and educate patients “virtually” by allowing remote information visualization [16,18,19,57,58,82,83]. Remote XR(VR)-based observation has been reported to be accurate, convenient, non-invasive, and time and specimen saving [84].

In telemedicine, the combination of wearable sensors, high-performance video, and HMD can be useful as diagnostic and follow-up support tools. Haddas et al. [82] reported that cameras and gait analysis platforms have described patterns of gait dysfunction caused by conditions such as myelopathy. In addition, patients who receive telemedicine for spine-related complaints have given high ratings for overall satisfaction and ease of use [85].

Although telemedicine spread rapidly during the COVID-19 pandemic, it has not yet been determined how best to perform an objective physical examination of the spine in the remote setting [85]. The limitations of telemedicine in performing a physical examination or collecting proper manual neurological findings in the field of spine medicine have been the biggest obstacle to adoption by spinal surgeons [85]. Guidelines have been developed for the implementation of virtual physical examinations, including specific testing methods for spine surgeons to assess the physical function, motor strength, and sensation [17,86,87,88]; however, these are still insufficient. In the future, research is needed to standardize remote control and improve the accuracy of remote diagnoses.

### 4.3. Surgery

There are increasing reports of hologram-based navigation surgery, which aims to improve existing surgical navigation by recreating the patient’s anatomy in 3D and superimposing it on the surgeon’s field of view (Figure 2). The development of XR technology with HMD [89] has contributed to hologram-based navigation. Moreover, Terander et al. [90] developed augmented reality surgical navigation (ARSN) for pedicle screw placement in the hybrid XR operating room. HMD visualization has solved the problem of the risk of attention shift, where the surgeon looks away from the surgical field to observe the monitor.

There have been numerous reports on proof of concept and surgical simulation with cadaveric, phantom, and animal models using XR technology: pedicle screw insertion (cervical [91], thoracic [92,93], thoracolumbar [94,95], lumbar [31,63,64,93,96,97,98,99,100,101]), cervical lateral mass screw insertion [32], vertebral body puncture [102,103,104], vertebroplasty (kyphoplasty) [105,106,107,108,109], percutaneous sacroiliac screw insertion [110,111,112], percutaneous lumbar discectomy [105,113,114,115], and facet joint injection [116,117,118]. However, few studies have evaluated the application of XR technology in clinical practice for pedicle screw placement [90,119], targeted cervical foraminotomy [120], osteotomy planning [121] and percutaneous intervention [122,123,124], or extradural and intradural tumor resection [125,126].

#### 4.3.1. Surgical Simulation

A summary of surgical simulation using hologram with XR technology in spine surgery is shown in Table 1.

Surgical simulations, which range from 3D computer environments to virtual simulations using HMD, are commonly used for training purposes and to assess competency in surgical skills [2,127]. Advances in XR technology for spinal surgery have recently gained traction in comparison to other surgical specialties. With the increase in MISS, the need for simulators to improve procedural skills and visuospatial awareness is important [76]. XR-technology-based simulation can provide residents and students with more immersive and realistic simulations based on actual surgical procedures, with a variety of sensory feedback, including a high-fidelity haptic system. XR-technology-based simulation not only helps residents develop and master their skills, but also helps maintain the skill levels of senior surgeons.

Surgical simulators using XR technology in orthopedics include arthroscopy simulators for shoulder, knee, and hip joint surgery; fracture fixation simulators; and drill simulators [39,127]. In spine surgery, surgical simulators include pedicle screw insertion, vertebroplasty, transvertebral anterior cervical foraminotomy, posterior cervical laminectomy and foraminotomy, percutaneous transforaminal endoscopic discectomy, lumbar puncture, and facet injection [76,77]. Lohre et al. [76] reviewed 38 studies that incorporated XR into MISS and reported that training using VR simulators outperformed traditional training methods in terms of both the knowledge and skills of learners in novice and expert surgeons.

Case-specific 3D holograms can provide young surgeons with a more precise image to understand surgical procedures in comparison to a surgery or anatomy textbook [128] (Figure 2A). Therefore, surgical simulation using case-specific 3D holograms can contribute to shortening the operative time and improving surgical safety.

#### 4.3.2. Surgical Navigation

A summary of surgical navigation using hologram with XR technology in spine surgery is shown in Table 2. XR-technology-based navigation in spine surgery has been reported to contribute to safety and accuracy, reduced operating time, reduced radiation exposure, and improved workload [111,129].

#### 4.3.3. Safety and Accuracy

The accuracy of percutaneous pedicle procedures, including screw placement, biopsy and vertebroplasty, has been the topic of concern that has received the most attention because the wrong approach to a pedicle can lead to nerve and vascular damage. Therefore, safety and accuracy are important parameters not only for skilled doctors but also for less skilled operators in an XR-technology-based navigation system. In fact, XR-technology-based navigation has the potential to better inform and assist surgeons in demanding and challenging operations that require a high degree of precision and accuracy. Regarding the safety and accuracy of XR-technology-based navigation, the Food and Drug Administration (FDA) has laid out strict parameters that these systems must meet for the approval of procedures such as pedicle screw placement in accordance with Standard Practice for Measurement of Positional Accuracy of Computer Assisted Surgical Systems (<3 mm screw tip deviation, <3° angular deviation) [7,130]. Terander et al. [90] showed an overall accuracy of 94.1% for pedicle screw placement, no severe misplacement, and a mean screw placement time of 5.2 min in the first prospective cohort study of pedicle screw placement with an XR-technology-based operating room based on a video system with four cameras. Subsequently, Burstorm et al. [131], based on the work of Terander et al., evaluated the feasibility and accuracy of a radiation-free XR-technology-based navigation system on two dead pigs and demonstrated a high degree of accuracy (1.7 mm accuracy at the entry point, 2.0 mm accuracy at the device tip, angular deviation of <2°, and insertion time of 195 s). In the future, it would be desirable to verify the results in cases with narrow cervical or thoracic pedicles, complex spinal deformities, or previous surgery [122]. These studies revealed the great potential of XR technology systems in MISS.

Carl et al. applied XR technology for epidural and intradural tumor resection, reporting high accuracy, with an average registration error of 1 mm [125,126]. Advanced visualization with XR technology, combined with additional views displayed on a screen near the surgical field, demonstrated the importance of 3D perception, including good depth-perception resulting in smooth hand–eye coordination [125,126]. The potential for the improvement of accuracy and safety of spine surgery procedures appears to be a significant benefit associated with adopting such technology.

#### 4.3.4. Reduced Operating Time and Radiation Exposure

The purpose of research into XR technology in spine surgery, besides accuracy and safety, is to reduce the operating time and avoid unnecessary radiation exposure among patients and operating theater staff. Particular attention should be paid to MISS surgery, which is associated with high radiation exposure [43,44]. Several studies have shown a significant reduction in operative time and radiation exposure when using XR technology in comparison to traditional surgical methods [68,69,119,122,125,126]. Theoretically, XR-technology-based navigation could improve the operative efficiency, especially in complex cases where surgeons are required to shift their attention back and forth between the patient and the fluoroscopy monitor all while directing or advancing an instrument. Since XR technology is still in its infancy and new technologies have a learning curve, further improvements can be expected as more cases are followed over time.

#### 4.3.5. Workload

Increased workload due to the learning curve of new technology and visual discomfort and fatigue caused by wearing an HMD for a long time are still problems because XR technology is still in the early stages. Since it is necessary to show that XR technology can also significantly reduce workloads, validated questionnaires to check the workload associated with XR technology have been developed, including the Surgical Task Load Index, System Usability score, and NASA-Task load Index (NASA Human Performance Research Group, 1987). Although these questionnaires have been used to evaluate the workload associated with the use of XR technology in spine surgery [132] and liver surgery [8], the studies have not been sufficient. Therefore, assessing the workload with XR technology in spine surgery is a future challenge.

### 4.4. Rehabilitation

The application of VR in medicine is not a new technology that has emerged in the 5G era; the technology that is applied to medicine can be traced back at least 20 years [16]. In terms of patient care, VR has been used as a distraction to relieve pain, known as “VR Analgesia”, which has been shown to have analgesic effects.

XR’s novelty and immersive nature have been proven to promote motivation, excitement, and task engagement, making virtual rehabilitation more effective than traditional rehabilitation [133]. VR technology has recently been implemented in various clinical settings, including physical, vocational, cognitive, and psychological rehabilitation [134]. The Biodex Balance System (BBS) is a reliable and objective tool for balance assessment and training [135]. Ibrahim et al. [136] objectively demonstrated the effectiveness of VR rehabilitation using the Biodex Balance System. Task-oriented training and treatment gamification also make rehabilitation enjoyable and motivate patients to carry out repetitive tasks. It has been widely adopted in many medical fields, especially in digital healthcare [133,134,136]. Therefore, XR-technology-based devices can improve analgesia, compliance, and the effectiveness of rehabilitation in spine medicine, including rehabilitation.

Investigations of VR-based rehabilitation in orthopedic rehabilitation have been carried out in analgesia and have investigated cervical [137,138,139], lumbar, shoulder, and knee [56,140] range of motion and motor function improvement. In addition, the XR-technology-based system has enabled patients to undergo standard rehabilitation at home without having to go to a hospital [141,142]. With the advent of an aging society, the number of patients with spinal diseases has been growing. In many countries, a shortage of human resources—such as physical therapists and caregivers—as well as the cost of home and outpatient care, have caused a mismatch between supply and demand in clinical practice [141]. The application of VR technology to additional processes in the field of rehabilitation may reduce the cost and time burden of rehabilitation for doctors and patients. However, this needs to be verified in the future.

## 5. Limitations, Future Directions, and Possibilities

XR technology in spine medicine is still associated with some limitations and challenges. First, it should be compared to investigate the cost-effectiveness (cost, time, comfort, learning curve, satisfaction, and clinical outcomes) because most studies that include substantial data on efficacy and efficiency have been lacking. In recent years, the quality of introductory XR-technology-based devices has continued to improve and the cost has declined, which may reduce the economic burden. Therefore, it will be important to standardize outcomes and identify key clinical parameters that will further assess the need for, benefits of, and economic relevance of this technology to promote its widespread adoption (e.g., the accuracy of collecting virtual physical findings in medical examinations, the accuracy of pedicle screw placement, and the reduced operating time and radiation exposure of surgical simulation/navigation). While higher costs may be a hindrance to implementation at present, the potential for improved accuracy, shorter operating times, and reduced radiation exposure would overcome the high initial cost of these systems. In Japan, hologram-based navigation for spinal surgery is covered by national medical and health insurance. To foster the widespread use and application of XR technology, the creation of diagnostic holograms should be covered by medical and health insurance, such as in cases where it is difficult to understand the anatomy. Second, XR-technology-based navigation has problems that are specific to early technologies: variation in accuracy, image delay, low image resolution, low brightness and contrast, re-registration and re-calibration, installation costs, adaptation to operating room workflow, and staff training. The biggest problem with 3D holograms is the mismatch between the virtual model and the real world, which needs to be further improved. This may be one of the reasons why 3D holograms, although used as guides, are not yet widely used as navigation devices in the surgical field [142]. In addition, the effect of the patient’s respiratory movement must be taken into account during thoracic spine surgery [90]. At present, XR operating rooms equipped with radiation-free XR-technology-based navigation systems have the combined benefit of high accuracy with no radiation exposure [90].

Most variation in accuracy and technology-related problems is expected to be resolved with the continued advancement of information technology, such as complex feedback processing with AI and machine learning. Image delays can occur in renderings that require extensive processing; normally, this is handled by compressing the polygonized patient organ model to reduce the processing load. Advances in HMD capabilities have improved the problem of visibility of models in the operating room. On the other hand, HMDs have been associated with side effects, such as nausea, headaches, dizziness, fatigue, and vision problems, as well as concerns about battery life, secure network access, and limited movement due to cords [39]. Validation for workload assessment is required to solve the ergonomic problems of HMD. The use of mobile batteries, simplification of volume rendering of models, and cordless HMDs are some other solutions. The potential for high costs and time-consuming training of staff can be a problem specific to new technologies. There is also a risk that inexperienced users may not be able to deal with system failure during surgery.

Third, especially in the elderly population, there are barriers such as digital literacy and access to telehealth technology. If this issue remains unaddressed, the widespread use of telemedicine services using XR technology may be limited to certain populations. On the other hand, as smart devices and video conferencing become more widespread, we believe that older patients will become more comfortable embracing telemedicine.

Fourth, it is possible that we are not prepared for the ethical challenges (privacy, electronic security, legal implications, etc.) associated with the use of virtual environments [143], especially in the collection of personal information through the web by wearable sensors or an HMD for spine medicine. It is possible that recording and sharing personal data may threaten the privacy of individuals. As clinicians adopt technology for service delivery and practice management, it is necessary to discuss ethical issues that may interfere with the assessment and treatment process, effectiveness, and even safety. Naturally, clinicians need to consider the ethical principles of beneficence (maximizing patient benefit) and nonmaleficence (avoiding harm) when conducting assessments and interventions using virtual reality [143].

Lastly, instead of a systematic review, we applied a narrative review methodology to discuss various technologies, including different perspectives and criticisms. This is because the methodology of the systematic review approach requires a more rigorous focus, which was not compatible with the purpose of this paper. In fact, previous reviews on XR technology in spine medicine have not conducted meta-analyses or adequate systematic reviews due to heterogeneity in the study design, outcome measures, and variability. On the other hand, the lack of a systematic methodology in this review makes it impossible to obtain the highest level of current evidence for conditions and technologies. Another limitation of a narrative review is that it cannot seek out all the relevant literature and cover the scientific literature unbiasedly; instead, it discusses pivotal articles known to the authors [133]. It should also be noted that ongoing research may soon make it obsolete.

## 6. Conclusions

This narrative review discussed factors that have promoted the progression of XR technology in spine surgery, including digital transformation, the COVID-19 pandemic, and MISS. In the field of spine medicine, XR technology has been introduced in areas of education, diagnoses, surgery, and rehabilitation, with remarkable results. Although XR technology in spine medicine still has some limitations and challenges, these can be solved by digital transformation. In the future, the introduction of XR technology to spine medicine will bring about disruptive changes in medical education, clinical diagnoses, doctor-patient communication, treatment, and rehabilitation, and promote the rapid development of medicine.

## Figures and Tables

**Figure 1 jcm-11-00470-f001:**
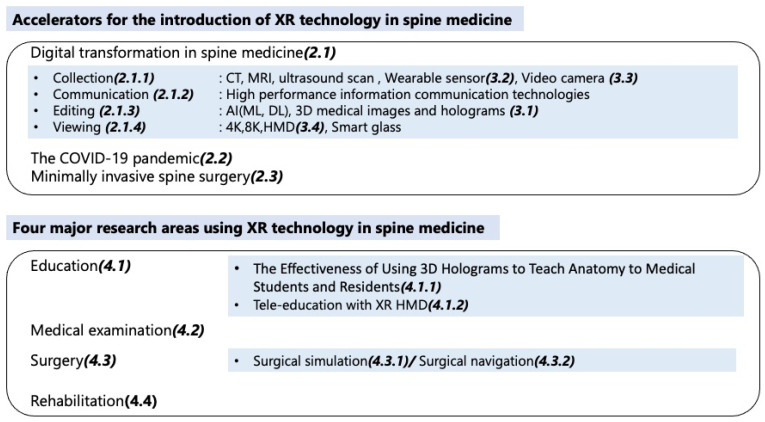
XR technology in spine medicine in contents of this manuscript. XR, extended reality; AI, artificial intelligence; ML, machine learning; DL, deep learning; HMD, head mounted display.

**Figure 2 jcm-11-00470-f002:**
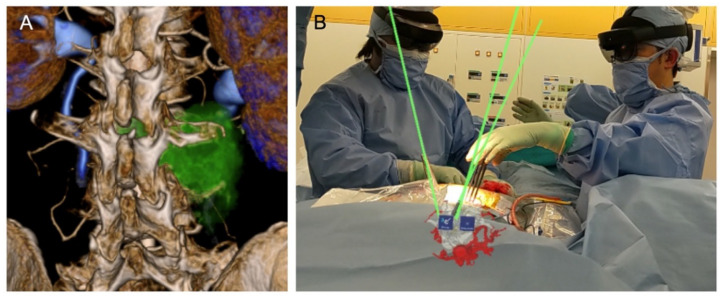
Three-dimensional medical images and holograms for preoperative simulation (**A**) and intraoperative surgical assistant (**B**). The green line was used as a guideline for the insertion of pedicle screws.

**Table 1 jcm-11-00470-t001:** A summary of surgical simulation with XR technology in spine surgery.

Study	Country	Procedure	VR/AR	Model	Simulator	Participant	Outcome
Hou et al., 2018 [91]	China	Cervical pedicle screw	VR-simulation	Phantom model, Cadaver model	Virtual Surgery Training System (VSTS)	Residents	Accuracy
Hou et al., 2018 [92]	China	Thoracic pedicle screw	VR-simulation	Phantom model, Cadaver model	Virtual Surgery Training System (VSTS)	Residents	Accuracy
Luciano et al., 2011 [93]	USA	Thoracic pedicle screw	AR-guide	Phantom model	ImmersiveTouch (San Francisco, CA, USA)	Residents	Accuracy
Xiang et al., 2015 [94]	China	Thoracolumbar pedicle screw	VR-simulation	Phantom model	Proprietary cross-platform simulator written in C++	Residents	Time
Xin et al., 2018 [95]	China	Thoracolumbar pedicle screw	VR-simulation	Phantom model, Cadaver model	Unspecified VR system, UG NX8.0, Seimens, Munich, Germany	Trainees	Time
Chitale et al., 2013 [96]	USA	Lumbar pedicle screw	AR-guide	Phantom model	Medtronic Surgical Technologies	Residents	Accuracy, Time
Gasco et al., 2014 [31]	USA	Lumbar pedicle screw	AR-guide	Cadaver model	ImmersiveTouch (San Francisco, CA, USA	Medical students	Accuracy
Gibby et al., 2019 [63]	USA	Lumbar pedicle screw	AR-guide	Phantom model	Microsoft HoloLens (Redmond, WA, USA), Novarad OpenSight (American Fork, UT, USA	Medical students, Trainees	Accuracy, Time
Liebmann et al., 2019 [64]	Switzerland	Lumbar pedicle screw	AR-guide	Phantom model	Microsoft HoloLens (Redmond, WA, USA)	Surgeon	Accuracy
Luciano et al., 2011 [93]	USA	Lumbar pedicle screw	AR-guide	Phantom model	ImmersiveTouch (San Francisco, CA, USA)	Trainees, Residents	Accuracy
Ma et al., 2017 [97]	China	Lumbar pedicle screw	AR-guide	Phantom model	Unspecified developed surgical navigation system	Surgeon	Accuracy
Molina et al., 2019 [98]	USA	Lumbar pedicle screw	AR-guide	Cadaver model	Unspecified developed surgical navigation system (AR)	2 Surgeons	Accuracy, Questionnaire
Molina et al., 2020 [99]	USA	Lumbar pedicle screw	AR-guide	Cadaver model	Unspecified developed surgical navigation system (AR)	2 Surgeons	Accuracy
Mostafa et al., 2017 [100]	Canada	Lumbar pedicle screw	VR-simulation	Phantom model	NeurosimVR, ImmersiveTouch (San Francisco, CA, USA)	Surgeons	Questionnaire
Rambani et al., 2014 [101]	UK	Lumbar pedicle screw	VR-simulation	Phantom model	Unspecified developed computer-assisted orthopedic training system	Trainees	Accuracy, Time, Radiation exposure
Gottschalk et al., 2015 [32]	USA	Cervical lateral mass screw	AR-guide	Cadaver model, Sawbone model	Stealth 3D Navigation Unit, Medtronic, Minneapolis, MN, USA	Residents	Accuracy
Fritz et al., 2013 [102]	USA	Lumbar vertebral body puncture	AR-guide	Cadaver model	Unspecified developed magnetic resonance (MR)-guided osseous biopsy	Radiologist	Accuracy, Time
U-Thainual et al., 2013 [103]	Canada	Lumbar vertebral body puncture	AR-guide	Phantom model	Unspecified developed MRI-guided musculoskeletal interventions Magnetic Resonance Image Overlay System (MR-IOS).	Operators	Accuracy, Time
Färber et al., 2009 [104]	Germany	Lumbar vertebral body puncture	VR-simulation	Phantom model	Sensable Phantom Premium 1.5	Medical students	Accuracy
Deib et al., 2018 [105]	USA	Vertebroplasty (Kyphoplasty)	AR-guide	Phantom model	Unspecified developed system	Operators	Accuracy, Time
Koch et al., 2019 [106]	Germany	Vertebroplasty (Kyphoplasty)	VR-simulation	Phantom model	VR vertebroplasty simulator	Operators	Questionnaire
Weigl et al., 2016 [107]	Germany	Vertebroplasty (Kyphoplasty)	VR-simulation	Phantom model	Novint Falcon (Novint Technologis, Inc., Albuquerque, NM, USA)	Surgeons	Radiation exposure, Workload (SURG-TLX scores (mental workload))
Wucherer et al., 2014 [108]	Germany	Vertebroplasty (Kyphoplasty)	VR-simulation	Phantom model	Novint Falcon (Novint Technologis, Inc., Albuquerque, NM, USA)	Surgeons	None
Wucherer et al., 2015 [109]	Germany	Vertebroplasty (Kyphoplasty)	VR-simulation	Phantom model	Novint Falcon (Novint Technologis, Inc., Albuquerque, NM, USA)	Surgeons	Radiation exposure, Workload (SURG-TLX scores (mental workload))
Dennler et al., 2021 [110]	Switzerland	Percutaneous sacroiliac screw insertion	AR-guide	Sawbone model	Unspecified developed system	Surgeons	Accuracy
Jeong et al., 2019 [111]	Korea	Percutaneous sacroiliac screw insertion	VR-simulation	Phantom model, Cadaver model	Unspecified developed system	Surgeons	Accuracy
Wang et al., 2016 [112]	China	Percutaneous sacroiliac screw insertion	AR-guide	Cadaver model	Unspecified developed system	Surgeons	Accuracy
Deib et al., 2018 [105]	USA	Lumbar percutaneous lumbar discectomy	AR-guide	Phantom model	Unspecified developed system	Operators	Accuracy, Time
Bisson et al., 2010 [113]	Canada	Lumbar percutaneous lumbar discectomy	VR-simulation	Phantom model	Unspecified developed system	Operators	Accuracy
Hu et al., 2017 [114]	China	Lumbar percutaneous lumbar discectomy	VR-simulation	Phantom model	Unspecified developed system	Operators	Time
Zhou et al., 2019 [115]	China	Lumbar percutaneous lumbar discectomy	VR-simulation	Phantom model	Unspecified developed system	Operators	Time
Moult et al., 2013 [116]	Canada	Lumbar facet joint injection	VR-simulation	Phantom model	Perk Tutor, SonixTouch US system with SonixGPS	Medical students	Accuracy, Time
Moore et al., 2009 [117]	Canada	Lumbar facet joint injection	VR-simulation, AR-guide	Phantom model	Unspecified developed system	Anesthetists	Accuracy
Yeo et al., 2011 [118]	Canada	Lumbar facet joint injection	VR-simulation	Phantom model	Perk Station (The Perk Lab, Queen’s University, Canada)	Medical students	Accuracy, Time

XR, extended reality; VR, virtual reality; AR, augmented reality.

**Table 2 jcm-11-00470-t002:** A summary of surgical navigation with XR technology in spine surgery.

Study	Country	Procedure	VR/AR	Simulator	Outcome
Elmi-Terander et al., 2019 [90]	Sweden	Lumbar pedicle screw	VR-simulation, AR-guide	Unspecified developed system—the ARSN system	Accuracy, time, clinical outcomes
Edström et al., 2019 [119]	Sweden	Lumbar pedicle screw	VR-simulation, AR-guide	Unspecified developed system—the ARSN system	Ratiation exposure, clinical outcomes
Umebayashi et al., 2018 [120]	Japan	Cervical foraminotomy	AR-guide	Medtronic StealthStation S7	Feasibility of intraoperative use
Kosterhon et al., 2017 [121]	Germany	Lumbar osteotomy planning	XR—volume rendered spine with VR preoperative planning and AR intraoperative workflow	Amira R, FEI Visualization Sciences Group, version 5.4.2, Mérignac Cedex, France	Feasibility of intraoperative use, clinical outcomes
Abe et al., 2013 [122]	Japan	Lumbar percutaneous intervention	AR-guide	Unspecified developed system-Virtual Protractor with Augmented Reality (VIPAR)	Accuracy
Wei et al., 2019 [123]	China	Lumbar percutaneous intervention	AR-guide	Baholo, Shanghai Front Computing Company, China; Medivi, Changzhou, China; Hololens, Microsoft, USA	Time (ope, radiation), clinical outcomes
Wu et al., 2014 [124]	China	Lumbar percutaneous intervention	AR-guide	Unspecified developed system	Accuracy, time (ope, radiation)
Carl et al., 2019 [125]	Germany	Extra- and intradural tumor resection (whole spine)	AR-guide	Unspecified developed system	Feasibility of intraoperative use
Carl et al., 2019 [126]	Germany	Extra- and intradural tumor resection (whole spine)	AR-guide	Unspecified developed system	Accuracy, ratiation exposure

XR, extended reality; VR, virtual reality; AR, augmented reality.

## Data Availability

Not applicable.

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
