# Peer review of "XR (Extended Reality: Virtual Reality, Augmented Reality, Mixed Reality) Technology in Spine Medicine: Status Quo and Quo Vadis"

_jcm, 2022, doi:10.3390/jcm11020470_

Round 1

Reviewer 1 Report

This is an interesting review paper about new technology in spine medicine. The paper is well designed and written. It would be of value to know efficacy and safety of the new technology.

Regarding the future directions, the author mentioned about cost-effectiveness, and the reviewer was also totally agreed with the authors. For spreading and applying this new technology in spinal medicine area, national medical and health insurance coverage is also important issue. The author would be better to mention briefly about the current status and future directions of national medical insurance coverage for this new technology in the world.

Author Response

Response to Reviewer 1

We thank the esteemed reviewer for their helpful comments.

>Regarding the future directions, the author mentioned about cost-effectiveness, and the reviewer was also totally agreed with the authors. For spreading and applying this new technology in spinal medicine area, national medical and health insurance coverage is also important issue. The author would be better to mention briefly about the current status and future directions of national medical insurance coverage for this new technology in the world.

Response: We have now added the following text to Line 575:

“In Japan, hologram-based navigation for spinal surgery is covered by national medical and health insurance. To foster the widespread use and application of XR technology, the creation of diagnostic holograms should be covered by medical and health insurance, such as in cases where it is difficult to understand the anatomy.”

Reviewer 2 Report

Title of the article:

The title should not contain abbreviations, eg XR (VR / AR / MR), please change it.

Abstract: Please precisely define the purpose of the article.
1. Introduction: Please explain the terms in detail: XR (VR / AR / MR).

2. What has accelerated the introduction of XR technology in spine medicine?
Please mention: In a study conducted in Poland in 2012-2013, an alarmingly high percentage of exceedances was found for AP / PA chest X-rays over the entire age range, and especially for younger children. The reason is the lack of standard procedures and practical shortcomings in meeting the requirements of Polish regulations and the EURATOM Directive. According to Hartmann et al, the combination of a clinical trial and the surface topography method results in a reduction in the number of radiographs, which should be welcomed in the context of radiation load. Prujis et al. estimated that the use of surface topography can eliminate up to 50% of radiography. Topographic measurements of the back surface partly meet these requirements and constitute an independent and promising alternative.

3. Technologies supporting the digital transformation of spine medicine:

Describe here the optoelectronic method for examining the spine Diers formetric III 4 D.

4 Three-dimensional spinal analysis comprises a combination of optical imaging techniques and digital data processing. This is a quick and non-contact photogrammetric measurement method allowing analysis of a patient's back and spine (Wilczyński J. Electromyographic activity of the erector spinae and convexity as well as concavity of spinal curvature in children. Children 2021, 8 (12), 1168 ; https://doi.org/10.3390/children8121168).

5. XR technology in spine medicine: education, medical examination, surgery, and 312 rehabilitation.
Please mention here the balancing platforms, such as the Biodex Balance System, on which, on the basis of biofeedback, patients suffering from, for example, Parkinson's are reghabited.

Conclusion:
 Colclusie should be precisely related to the title of the article and its purpose.

Author Response

Response to Reviewer 3

We thank the esteemed reviewer for their helpful comments. We hope we have improved the quality of our work through these changes. The individual remarks are explained below.

>The title should not contain abbreviations, eg XR (VR / AR / MR), please change it.

Response: We have now revised the title as suggested: “XR (Extended Reality: Virtual Reality, Augmented Reality, Mixed Reality) technology in spine medicine: Status Quo and Quo Vadis”

>Abstract: Please precisely define the purpose of the article.

Response: The purpose of the paper was unclear, so we revised the text as follows in Line 26 of the Abstract and Line 94 of the Introduction:

“The purpose of this narrative review is to describe the accelerators of XR (VR, AR, MR) technology in spine medicine and then to provide a comprehensive review of the use of XR technology in spine medicine, including surgery, consultation, education and rehabilitation, as well as to identify its limitations and future perspectives (Status Quo and Quo Vadis).”

>1. Introduction: Please explain the terms in detail: XR (VR / AR / MR).

Response: The following explanations of terms have been added:

VR: Line 41 - It is widely used to reproduce the human body structure, pathophysiology, and clinical scenes with a sense of realism

AR: Line 47 - In practice, computer-generated images are overlaid on real-world images and displayed on video projectors, computers, or tablets.

MR: Line 49 - MR, a hybrid of AR and VR, is the result of blending the physical world with the digital world [5,6] and has recently garnered attention, mitigating the limitations of VR’s exclusion of the real-world environment and AR’s inability to interact with 3-dimensional (3D) data packet

XR:Line 59- XR is a general term encompassing VR, AR, and MR.

>2. What has accelerated the introduction of XR technology in spine medicine?
Please mention: In a study conducted in Poland in 2012-2013, an alarmingly high percentage of exceedances was found for AP / PA chest X-rays over the entire age range, and especially for younger children. The reason is the lack of standard procedures and practical shortcomings in meeting the requirements of Polish regulations and the EURATOM Directive. According to Hartmann et al, the combination of a clinical trial and the surface topography method results in a reduction in the number of radiographs, which should be welcomed in the context of radiation load. Prujis et al. estimated that the use of surface topography can eliminate up to 50% of radiography. Topographic measurements of the back surface partly meet these requirements and constitute an independent and promising alternative.

Response: As suggested, the following text has been added to Line 129 and 135:

Line 129: “and 4) development of surface topography methods in pediatric scoliosis.”

Line 134: The use of surface topography in pediatric scoliosis has been welcomed for its accuracy and reduced radiation exposure [23].

>3. Technologies supporting the digital transformation of spine medicine:

Describe here the optoelectronic method for examining the spine Diers formetric III 4 D.

Response: We completely agree with the reviewer concerning the usefulness of surface topography. We have added the following new section:

3.1.3. Surface topography in spinal posture

>4 Three-dimensional spinal analysis comprises a combination of optical imaging techniques and digital data processing. This is a quick and non-contact photogrammetric measurement method allowing analysis of a patient’s back and spine (WilczyÅ„ski J. Electromyographic activity of the erector spinae and convexity as well as concavity of spinal curvature in children. Children 2021, 8 (12), 1168 ; https://doi.org/10.3390/children8121168).

Response: Surface topography is discussed in section 3.1.3 mentioned above. The paper the reviewer suggested by WilczyÅ„ski et al. is not relevant to our report, as that paper appears to concern EMG, and we were unable to find it, as it is not on Pubmed.

>5. XR technology in spine medicine: education, medical examination, surgery, and 312 rehabilitation. Please mention here the balancing platforms, such as the Biodex Balance System, on which, on the basis of biofeedback, patients suffering from, for example, Parkinson's are reghabited.

Response: We appreciate your information on balancing platforms, such as the Biodex Balance System. Line 635 now mentions the Biodex Balance System.

The Biodex Balance System (BBS) is a reliable and objective tool for balance assessment and training [157]. Ibrahim et al. [158] objectively demonstrated the effectiveness of VR rehabilitation using the Biodex Balance System.

>Conclusion:
Colclusie should be precisely related to the title of the article and its purpose.

*********

Response: Title: XR (Extended Reality: Virtual Reality, Augmented Reality, Mixed Reality) technology in spine medicine: Status Quo and Quo Vadis

“The purpose of this narrative review is to identify the accelerators of XR (VR, AR, MR) technology in spine medicine and then to perform a comprehensive review of the use of XR technology in spine medicine, including surgery, consultation, education and rehabilitation, as well as to identify its limitations and future perspectives (Status Quo and Quo Vadis).”

**********

Corresponding to the title and purpose, the following is a brief description of the project.

“This narrative review discussed factors that have promoted the progression of XR technology in spine surgery, including digital transformation, the COVID-19 pandemic, and MISS. In the field of spine medicine, XR technology has been introduced in areas of education, diagnoses, surgery and rehabilitation, with remarkable results. Although XR technology in spine medicine still has some limitations and challenges, these can be solved by digital transformation. In the future, the introduction of XR technology to spine medicine will bring about disruptive changes in medical education, clinical diagnoses, doctor-patient communication, treatment, and rehabilitation and promote the rapid development of medicine.”

Round 2

Reviewer 2 Report

After the corrections, I accept the article